# Informed Local Smoothing in 3D Implicit Geological Modeling

**Jan von Harten** [1,*] **, Miguel de la Varga** [2] **, Michael Hillier** [3] **and Florian Wellmann** [1,2]

1 Computational Geoscience and Reservoir Engineering, RWTH Aachen University, 52062 Aachen, Germany; wellmann@aices.rwth-aachen.de
2 Terranigma Solutions GmbH, 52072 Aachen, Germany; miguel@terranigma-solutions.com
3 Geological Survey of Canada, Ottawa, ON K1A 0E8, Canada; michael.hillier@canada.ca
* Correspondence: vonharten@cgre.rwth-aachen.de

**Abstract:** Geological models are commonly used to represent geological structures in 3D space. A wide range of methods exists to create these models, with much scientific work focusing recently on implicit representation methods, which perform an interpolation of a three-dimensional field where the relevant boundaries are then isosurfaces in this field. However, this method has well-known problems with inhomogeneous data distributions: if regions with densely sampled data points exist, modeling artifacts are common. We present here an approach to overcome this deficiency through a combination of an implicit interpolation algorithm with a local smoothing approach. The approach is based on the concepts of nugget effect and filtered kriging known from conventional geostatistics. It reduces the impact of regularly occurring modeling artifacts that result from data uncertainty and data configuration and additionally aims to improve model robustness for scale-dependent fit-for-purpose modeling. Local smoothing can either be manually adjusted, inferred from quantified uncertainties associated with input data or derived automatically from data configuration. The application for different datasets with varying configuration and noise is presented for a low complexity geologic model. The results show that the approach enables a reduction of artifacts, but may require a careful choice of parameter settings for very inhomogeneous data sets.

**Keywords:** 3D modeling; implicit modeling; geomodeling; geostatistics; kriging; nugget effect; kernel density estimation





## 1. Introduction

In many geoscientific problems, a representation of structures and boundaries in the subsurface forms the basis for scientific and economic endeavors (see [1] for a recent overview). The consideration of 2D manifolds in 3D space relates to the underlying geological concept that abrupt events in geological history are today often present in the form of significant changes in rock properties, which are then observed as distinct boundaries. Similarly, tectonic events often result in localized deformation zones which, in the case of brittle deformation, leads to the development of faults and fault networks, which can often be approximated by 2D manifolds for many purposes.

Depending on their purpose, models need to be able to adequately represent specific structures with varying degrees of accuracy over defined scales. While the desired resolution might be defined by the posed problem, available input data generally spans over a wide range of scales, from high resolution small-scale (cm) borehole data to low-resolution large-scale geophysical measurements (km). One of the main challenges of creating a geologic model is to integrate localized information of varying quality at different scales into a single fit-for-purpose model [2].

When we talk about geologic modeling, in general we mean a representation of 3D surfaces in space that represent abrupt changes of rock properties, often corresponding to geological events in history. These surfaces can either be modeled explicitly as three-dimensional surface meshes or implicitly as iso-surfaces of a continuous scalar field interpolated over the full domain space [1].

A widely used implicit method for 3D structural geologic modeling is the potential field method [3]. It combines information on geologic contact points (surface points) and orientation data in a universal cokriging system to define a scalar field [3–6]. Iso-surfaces of this scalar field can be extracted as explicit representations of geologic boundaries [7]. Multiple surfaces can thus be implicitly represented by a single scalar field. Unconformable relations are modeled by combinations of multiple scalar fields, while faults are represented by drift functions in the cokriging system [4,6]. The method is implemented in a range of software packages and has been successfully applied in various case studies to investigate crustal architectures (see, e.g., in [8–10]), geothermal and hydrogeological settings (e.g., [11,12]) and mineral systems (see, e.g., in [13–15]).

The geostatistical concept that forms the theoretical basis of the potential field method is the spatial interpolation method called kriging. It was developed in the 1950s [16] and formalized in the 1960s [17] and is thus well described in the literature [5,18,19]. While the application to structural geologic modeling [3] is a more recent development, taking advantage of the vast available literature from other fields supports finding novel solutions to existing problems of the potential field method.

A common problem reported by practitioners is the tendency of the method to create unrealistic and undesirable, mostly circular modeling artifacts (e.g., [20,21]). These artifacts are a result of the implicit, data-based modeling process that does not incorporate geologic expertise in its pure form. We identified four main solutions that have been applied by users to tackle these problems and improve models created with the potential field method (compare Figure 1):

1. Parameter optimization in the interpolation step. This means predominantly adjusting the used model of spatial correlation to improve the model. This can either be the theoretical covariance model that is used, or its parameters, especially the range [3,22].
2. Adding additional constraints as input data. This can include manually added support points based on geologic knowledge or the introduction of additional data types such as inequality constraints [23] or second-order field derivatives [24].
3. Postprocessing methods have been proposed to validate models after computation. Typically, a set of realizations is created and only viable models are accepted. Model validation can, for example, be based on geologic knowledge [25], topology [26] or some form of test data [21].
4. Preprocessing: Kriging results are largely dependent on the original data configuration. Contradicting data, data strongly varying over different scales and unevenly spaced data can lead to modeling artifacts. Proper cleaning, but also manual selection of used data is often required to achieve acceptable results [19].

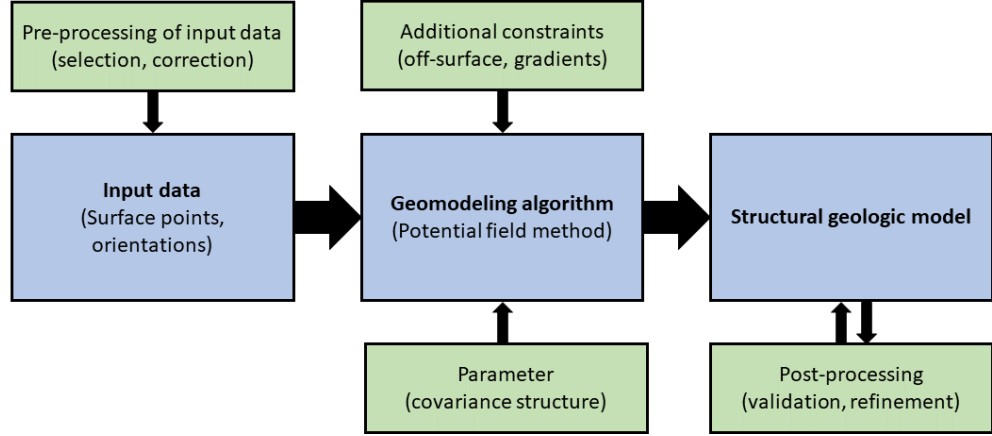

**Figure 1.** Typical geomodeling workflow using the potential field method (blue) and measures taken to improve models at different steps of the workflow (green).

The optimization of the applied covariance function (1) cannot reliably solve the occurrence of geologically unreasonable artifacts, while the addition of new constraints (2) is often very specific to certain modeling questions and available data and requires advanced knowledge of the algorithm and its implementation. Model validation in postprocessing (3) has proven useful, but requires a solid and automatable validation strategy as well as a suitable amount of robust model realizations. Finally, careful preprocessing of input data can lead to tedious manual work, while also compromising the reproducibility of the workflow.

We suggest here an automatable approach combining elements of (1) and (4) above by informing the methods parameters, specifically a portion of the nugget effect of the covariance function, with either original information or derived parameters of the original input data and its configuration.

In geostatistics, the nugget effect describes the behavior of a model of spatial correlation at small (zero) lag distances. There are two main sources of the nugget effect: (a) a small-scale variation of the observed phenomenon not captured by the dominant covariance structure or (b) measurement errors of the original input data [18,19]. Both effects equally apply to geomodeling, where the modeled phenomenon are the geometric surface boundaries.

In classical statistics the term heteroscedasticity describes the the variation of statistical dispersion of a random variable. In the framework of geostatistics, assumptions of stationarity of statistical moments are generally made to substitute multiple replications with measurements in space, related via a model of spatial correlation [18]. First-order stationarity refers to a constant mean and second-order stationarity to a covariance dependent only on the distance vector between two points [5,18,19]. Several methods exist to tackle non-stationarity of the mean in kriging applications: Most commonly used are universal kriging [27], kriging with External Drift (KED) [18] and regression kriging (RK) [28].

Modeling second-order non-stationary fields has gotten more attention recently. In the context of geosciences, varying methods have been proposed to relax the assumption of second-order stationarity by applying a non-stationary covariance structure [29–31]. Other suggested methods alter the diagonal entries of the covariance matrix, analogue to the concept of classical heteroscedasticity, with additional information of the variances of the modeled process [32,33]. This is equivalent to locally changing portion of the nugget effect attributed to measurement error and has also been named kriging with known measurement error or filtered kriging [18,34,35]. Note that this does not violate the assumption of second-order stationarity, assuming that the measurement error free portion of the covariance is still translation invariant [18].

We apply this approach to geomodeling with the potential field method by changing the diagonal covariance entries of the surface point cokriging submatrix to achieve local smoothing of modeled surfaces. This allows us to account for (a) known uncertainties of different input data types and (b) small-scale variation captured by highly localized dense sampling that can not be resolved in a large scale model (compare Figure 2). The local nugget variation can be informed either by quantified uncertainties of the input surface data or by parameters of the input data configuration. As this method operates on the scalar field, not the real scale of the model, local nugget variance has to be scaled relative to the scalar field.

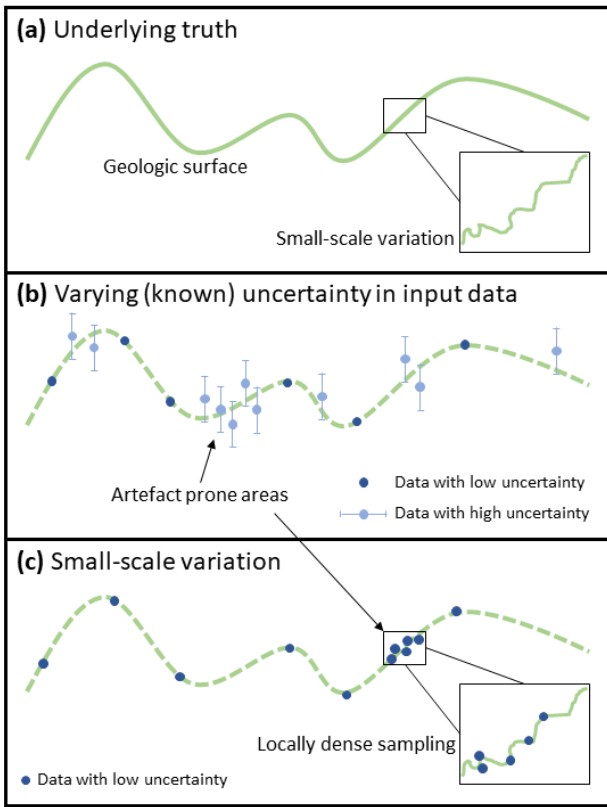

**Figure 2.** Modeling problems tackled by informed local smoothing: (**a**) Underlying truth with geologic surface. (**b**) Input data with varying quantified uncertainties. (**c**) Locally dense sampling of small-scale variation below desired model resolution.

The suggested approach reduces the accuracy of the resulting model, meaning its fit to data points, locally. While this is generally an undesirable behavior in classical geostatistics, we often require robust and geologically reasonable geostructural models. The exact fit to data points is less relevant, especially when measurements feature uncertainties or cover a range of scales at a higher resolution than needed for the purpose of the model.

In this paper, we will revisit the theoretical background of kriging, the nugget effect and the potential field method. The application of a local nugget effect in traditional 1D kriging applications is used to showcase the local smoothing effect of this approach on interpolation results. We present three approaches to inform local smoothing using manual, semi-automated and fully automated methods depending on available data. Finally, a simplified geological model with varying input data configurations is shown to illustrate the effect of local smoothing in 3D structural geomodeling.

## 2. Materials and Methods

In the following section, we will briefly present the concepts and relevant equations of kriging and the expansion to implicit geomodeling. The theory and role of the nugget effect in these geostatistical frameworks will be shown. We present a method to apply localized smoothing to input data points based on the diagonal of the kriging matrices. In a last step, we will introduce manual, semi-automated and fully automated procedures to inform the local smoothing, depending on the posed problem and the available data.

### 2.1. Ordinary Kriging

The potential field method used here is a kriging-based method. In order to illustrate the effect of smoothing we first establish a classical ordinary kriging system. Ordinary kriging is the most commonly used form of kriging, based only on the two assumptions that (1) the mean $\mu$ of the random function is unknown but constant and (2) that the we are

able to infer a valid variogram function from available data [18]. The predictor of ordinary kriging $Z^*_{\omega_{OK}}(\mathbf{x}_0)$ of the value of a random function $Z(\mathbf{x})$ at location $\mathbf{x}_0$ is defined as

$$Z^*_{\omega_{OK}}(\mathbf{x}_0) := \sum_{i=1}^{n} \omega_i Z(\mathbf{x}_i) \tag{1}$$

for a set of sampled data locations $\mathbf{x}_1, \mathbf{x}_2, \ldots, \mathbf{x}_n$ [17–19]:

The ordinary kriging system as first presented by Matheron [17] represents the resulting equations when optimized for minimal estimation variances $\delta_E^2$ to determine the weights $\omega_{OK}$. Minimization is achieved by defining a function dependent on these variances and a term containing the *Lagrange multiplier* and setting its partial derivatives in respect to the weights to zero. Compare Wackernagel [18] and Webster and Oliver [19] for a complete derivation. The set of resulting equations in matrix formulation is

$$\begin{bmatrix} C(\mathbf{x}_1 - \mathbf{x}_1) & \cdots & C(\mathbf{x}_1 - \mathbf{x}_i) & \cdots & C(\mathbf{x}_1 - \mathbf{x}_n) & 1 \\ \vdots & \ddots & \vdots & \ddots & \vdots & \vdots \\ C(\mathbf{x}_i - \mathbf{x}_1) & \cdots & C(\mathbf{x}_i - \mathbf{x}_i) & \cdots & C(\mathbf{x}_i - \mathbf{x}_n) & 1 \\ \vdots & \ddots & \vdots & \ddots & \vdots & \vdots \\ C(\mathbf{x}_n - \mathbf{x}_1) & \cdots & C(\mathbf{x}_n - \mathbf{x}_i) & \cdots & C(\mathbf{x}_n - \mathbf{x}_n) & 1 \\ 1 & \cdots & 1 & \cdots & 1 & 0 \end{bmatrix} \begin{bmatrix} \omega_1^{OK} \\ \vdots \\ \omega_i^{OK} \\ \vdots \\ \omega_n^{OK} \\ -\lambda_{OK} \end{bmatrix} = \begin{bmatrix} C(\mathbf{x}_1 - \mathbf{x}_0) \\ \vdots \\ C(\mathbf{x}_i - \mathbf{x}_0) \\ \vdots \\ C(\mathbf{x}_n - \mathbf{x}_0) \\ 1 \end{bmatrix} \tag{2}$$

with $\lambda$ denoting the lagrange multiplier and $C$ a valid covariance function. Weights obtained from solving these equations are used in the ordinary kriging predictor (Equation (1)) to calculate an estimate at any given position $\mathbf{x}_0$.

Note that we present here the common notation of ordinary kriging with a covariance $C$, as this is often used in implementations of the potential field method [3,6]. It can easily be rewritten to be used with a variogram function $\gamma$ following the relationship, where $\mathbf{h}$ is the lag distance and $C(\mathbf{0})$ is the total variance of the process:

$$\gamma(\mathbf{h}) = C(\mathbf{0}) - C(\mathbf{h}) \tag{3}$$

### 2.2. Nugget Effect and Filtered Kriging

The nugget effect in a geostatistical framework is a property of the model of spatial correlation. There are two inherently different ways to view the nugget effect. Traditionally, we assume that our measurements are exact, but there are small-scale variations in the modeled phenomenon that are not captured by the dominating spatial correlation model (e.g., the name giving gold nugget) [18,19]. This is done by introducing a discontinuous origin to the theoretical model of spatial correlation. In case of a variogram this means that the semivariance at lag distance zero is defined to be zero and thus original measurements are exactly honored in kriging interpolation. This pure nugget effect covariance model, where $b$ is the total variance of the process, is then defined as [18]

$$C^{nug} := \begin{cases} b, & \text{for } |h| = 0 \\ 0, & \text{for } |h| > 0 \end{cases} \tag{4}$$

Alternatively, the nugget effect can be viewed as measurement error or noise [18]. In this case, the semivariance at lag distance zero can have non-zero values and thus measurements are not necessarily honored exactly. Applied to kriging this means that we can have interpolated values different from original observations at measurement locations, leading to the term filtered kriging [34] or kriging with known measurement error [18]. In practice this can be implemented by using a valid covariance function with zero nugget effect and subsequently adding a noise variance to the diagonal of the kriging matrix.

In Figure 3 we illustrated the difference between both approaches in a 1D interpolation example. Note that in reality, a combination of both effects, small-scale variation and measurement error, should be considered.

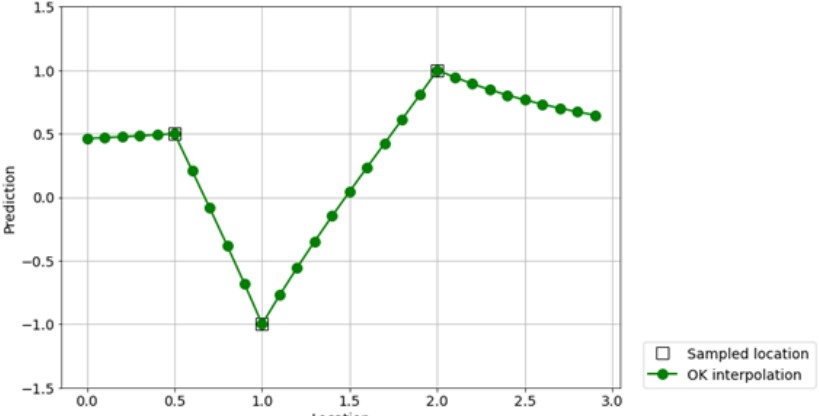

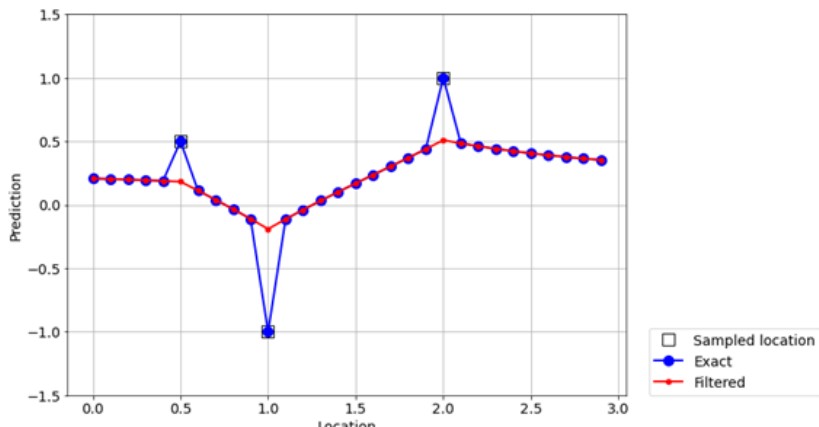

**Figure 3.** 1D Ordinary kriging interpolation using a exponential variogram model with range = 1 and sill = 1. Dots indicate discretized locations of interpolation. (**a**) Without nugget effect. (**b**) Nugget effect of 0.5. Blue: Exact nugget effect (small-scale variation); Red: Filtered kriging (measurement error). Adapted after K. Krivoruchko, A. Gribov, J. M. Ver Hoef [34].

It can be seen that a traditional nugget effect can lead to discontinuities in the interpolation at data locations depending on discretization (Figure 3b, blue line), while filtered kriging (Figure 3b, red line) leads to smooth results, but measurements are not honored and kriging ceases to be an exact interpolator. This means that the interpolation result at a sampled location does not necessarily equal the measurement at this location [18,19].

*2.3. Local Smoothing*

As described above the nugget effect in its traditional form is part of the nested structure of a model of spatial correlation, either a variogram or covariance model. In typical geostatistical applications it is modeled together with the complete structure and thus affects the derived theoretical model, and, subsequently the kriging interpolation, globally. In this work, we only consider the measurement error portion of the nugget effect, allowing us to only adjust the behavior at zero lag distance. In practice this is achieved by manipulating the diagonal of the redundancy kriging matrix (compare [18,33]):

$$\begin{bmatrix} C(\mathbf{x}_1 - \mathbf{x}_1) + \sigma_1^2 & \cdots & C(\mathbf{x}_1 - \mathbf{x}_i) & \cdots & C(\mathbf{x}_1 - \mathbf{x}_n) & 1 \\ \vdots & \ddots & \vdots & \ddots & \vdots & \vdots \\ C(\mathbf{x}_i - \mathbf{x}_1) & \cdots & C(\mathbf{x}_i - \mathbf{x}_i) + \sigma_i^2 & \cdots & C(\mathbf{x}_i - \mathbf{x}_n) & 1 \\ \vdots & \ddots & \vdots & \ddots & \vdots & \vdots \\ C(\mathbf{x}_n - \mathbf{x}_1) & \cdots & C(\mathbf{x}_n - \mathbf{x}_i) & \cdots & C(\mathbf{x}_n - \mathbf{x}_n) + \sigma_n^2 & 1 \\ 1 & \cdots & 1 & \cdots & 1 & 0 \end{bmatrix} \tag{5}$$

As this diagonal directly relates to the input data points, a different value can be assigned per input data point, leading to a heterogeneous variance distribution [35], comparable to statistical heteroscedasticity. In order to avoid confusion around the terminology of nugget effect [34], filtered kriging [35] and kriging with known measurement error [18] we will refer to this this local application of the measurement error portion of the nugget effect as local smoothing in the context of geomodeling.

Figure 4 shows the effect of local smoothing in 1D using the ordinary kriging system on a synthetic nine data point configuration. Note how it is possible to vary applied smoothing per data point, in Figure 4a with a smoothing of either 0 or 0.5, in Figure 4b with gradually increasing values from 0 to 1.6 as a function of measurement location.

**(a)** Local smoothing with manually fixed values

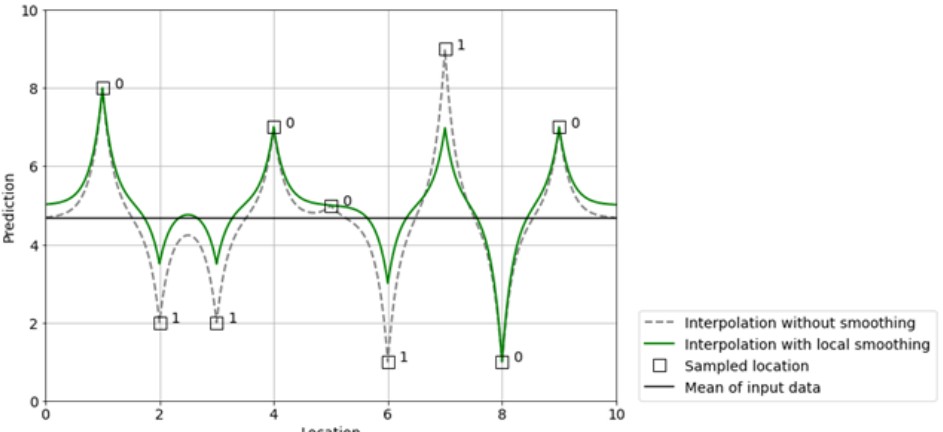

**(b)** Local smoothing with gradually increasing values

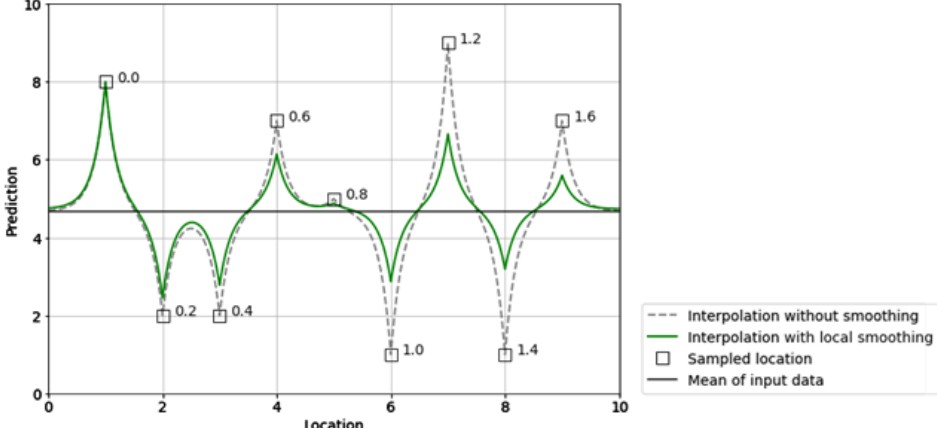

**Figure 4.** Application of local smoothing to 1D Ordinary kriging interpolation using an exponential model with range = 0.2 and sill = 1. Assigned local smoothing values noted next to sampled data locations. (**a**) Manually fixed local smoothing (either 0 or 1). (**b**) Local smoothing gradually increasing as function of location (from left (0) to right (1.6)).

### 2.4. Application in Geomodeling

We now transfer the concept of local smoothing to geomodeling with the potential field method. In the potential field method, two types of input data, namely, surface points and orientations, are used to interpolate a dimensionless scalar field that is implicitly representing multiple conformal geologic units. In a geological sense, surface points, meaning locations of abrupt changes in subsurface properties, in practice generally boundaries of geologic units, can be considered as time data. The resulting scalar field can be interpreted as a representation of relative age. Orientations are the gradient data of this scalar field. A universal cokriging system of the following form is used to combine these types of information [3]:

$$
\begin{bmatrix}
\mathbf{C}_{\delta Z/\delta u,\delta Z/\delta v} & \mathbf{C}_{\delta Z/\delta u,Z} & \mathbf{U}_{\delta Z/\delta u,Z} \\
\mathbf{C}_{Z,\delta Z/\delta u} & \mathbf{C}_{Z,Z} & \mathbf{U}_{Z} \\
\mathbf{U}'_{\delta Z/\delta u,Z} & \mathbf{U}'_{Z} & 0
\end{bmatrix}
\begin{bmatrix}
\omega_{\delta Z/\delta u,\delta Z/\delta v} & \omega_{\delta Z/\delta u,Z} \\
\omega_{Z,\delta Z/\delta u} & \omega_{Z,Z} \\
\mu_{\delta u} & \mu_{u}
\end{bmatrix}
=
\begin{bmatrix}
\mathbf{c}_{\delta Z/\delta u,\delta Z/\delta v} & \mathbf{c}_{\delta Z/\delta u,Z} \\
\mathbf{c}_{Z,\delta Z/\delta u} & \mathbf{c}_{Z,Z} \\
f_{10} & f_{20}
\end{bmatrix}
\tag{6}
$$

where $\mathbf{C}_{\delta Z/\delta u,\delta Z/\delta v}$ is the gradient covariance matrix, $\mathbf{C}_{Z,Z}$ is the surface point covariance matrix and the off-diagonal entries contain cross covariances and drift functions from universal kriging. Analogous to ordinary kriging, $\omega$ refers to the desired weights, $\lambda$ contains constant parameters of the estimated drift, $c$ are the covariances and cross-covariances between input points (surface and orientations) and the target location and $f_{10}$ and $f_{20}$ are the gradient of the universal drift function. We refer to Lajaunie et al. [3] and de La Varga et al. [6] for a full derivation and explanation of this system. Complex geologic features can be modeled either by introducing drift functions for faults or by combining multiple scalar fields for unconformities [4]. Surface meshes representing geologic boundaries can be extracted from the scalar field for visualization and postprocessing using a marching cube algorithm [7,36].

Analogue to the ordinary kriging example, local smoothing can be applied to the universal cokriging system. In this work we focus on surface point data and thus only manipulate the diagonal of the submatrix $\mathbf{C}_{Z,Z}$ containing surface data covariances, while leaving the other parts of the system untouched.

Note that the covariance model used in the universal cokriging remains unchanged and the assumption of second-order stationarity remains valid [18] as heteroscedasticity is only introduced at measurement locations. We now allow interpolated surfaces (or in general the scalar field) to deviate from given input surface data points in a controlled fashion. In this scenario, the potential field method ceases to be an exact interpolator.

### 2.5. Informing Local Smoothing

We established that we can locally smooth a geomodel based on a smoothing parameter defined for each surface input data point and why we are interested in that. The question remains how we can determine reasonable values for this parameter. In the following section we present a manual, semi-automated and automated method to inform the local smoothing. Note that combinations of these methods can easily be implemented to improve model control.

#### 2.5.1. Manually Informed

While not the main focus of this work, we want to emphasize that local smoothing can be used to manually adjust a geomodel. After model computation, undesirable artifacts that are tied to specific areas of challenging data configuration, such as contradicting information or high data density, can be assigned higher local smoothing. This is a mixture of pre- and postprocessing as shown in Figure 1, but offers a rather intuitive single parameter that can be adjusted manually in an iterative process.

#### 2.5.2. Data Informed

Geomodels are often created based on data from different sources [1]. This includes borehole, surface and geophysical data from multiple campaigns, often sampled over a

long period of time with strongly varying quality. Typically data is consolidated, evaluated and selected prior to geomodeling and sometimes has to be iteratively adapted if model results fail quality checks. This leads to an important differentiation in reproducibility of workflows: while the interpolation step is generally easily reproducible with provided algorithms, parameters and materials, preprocessing can involve tedious manual cleaning of data that is not always well documented (compare for example [37,38] and leads to practically irreproducible models.

We propose here that part of the data preprocessing can be substituted by assigning meaningful local smoothing values based on data type and source. These values can be inferred from (a) reported measurement errors for specific techniques and/or campaigns or (b) analysis and quantification of interpretation uncertainties e.g., in borehole data. We argue that this allows to semi-automate the process of data processing and selection and encourages the development of fully reproducible workflows that incorporate all available data sources.

### 2.5.3. Data Configuration-Informed

Data configuration has been reported to have a strong influence on kriging results [19]. In best case scenarios, this can be mitigated by smart sampling design, generally meaning regularly spaced measurements that cover the full model domain. In reality, geologic data is often highly localized, leading to areas of high data density for example along boreholes, at available outcrops or along seismic sections. This is one of the main reasons for modeling artifacts in the potential field method. Localized data might contradict the large-scale model, either because of small-scale variation or because of high variance due to measurement errors.

We suggest to use the well-known concept of Kernel density estimation (KDE) to infer local smoothing values. KDE is a nonparametric method from statistics to estimate probability density functions of random variables. For a full description of the approach see Scott and Sain [39] or Silverman [40]. In the context of input data configuration for geomodeling it provides a relative measure of data density. This way we can apply high local smoothing values to areas of high data density that are especially prone to lead to modeling artifacts, while exactly honoring isolated data points in areas where data density is low with low local smoothing values. For the examples in this work a gaussian kernel with automated bandwidth selection following Scott's rule [39,41] is used to estimate KDE. This provides an easily comparable relative measure of data density, but it has to be noted that other kernels or bandwidths might improve results for specific data configurations [42].

Note that this is distinctively different from the redundancy term in the kriging equations which reduces the overall weight of clustered points on the kriging interpolation [18]. The KDE informed local smoothing allows the interpolation to create smooth surfaces within point clusters by not following input data points exactly.

### 2.6. Scaling

The scalar field interpolated by the potential field method is a dimensionless field, its values only bear relative information of unit thickness and time in a geological sense [3]. Local smoothing values, independent of the way they were inferred, thus need to be scaled relative to this scalar field in order to produce reasonable results. In general, scaling requires a mapping between scalar field a real space. In case of low complexity models we can deduce this relation from a preliminary model result, comparing unit thickness and scalar field values. For higher complexity models, this approach is only valid if units feature constant thicknesses.

In case of KDE-informed local smoothing, scaling cannot be exactly defined, as both the scalar field and KDE are relative properties. For the GemPy implementation we suggest normalizing the KDE and using the resulting values between zero and one as local smoothing in the cokriging system. This has proven to work well with low-complexity

models, an iterative adjustment for high-complexity models based on the results might be necessary.

## 3. Results

In this section, we will show the application of informed local smoothing in geomodeling using a synthetic modeling example with varying data configuration.

### 3.1. Model with Regularly Spaced Data

We introduce a simple well-constrained model of a folded geologic structure. Data points to construct the model, while sparse, are regularly spaced and cover the whole model domain. A total of of 36 points are used to constrain the top and bottom of a single unit of interest. Each fold limb is defined by nine input data points per surface. Surface points are spaced 400 arbitrary units apart in y direction and 200 units apart in x direction, leading to a spacing along the hinge of roughly 283 units. Only a single orientation pointing upwards was added to the fold hinge for both modeled surfaces. The model was created with the potential field method in GemPy using a standard cubic covariance function. Figure 5 shows the input data configuration as well as the resulting 3D surface model. The extent of the model is $1000 \times 1000 \times 1000$ units computed with a resolution of $50 \times 50 \times 50$, corresponding grid spacing is thus 20 units.

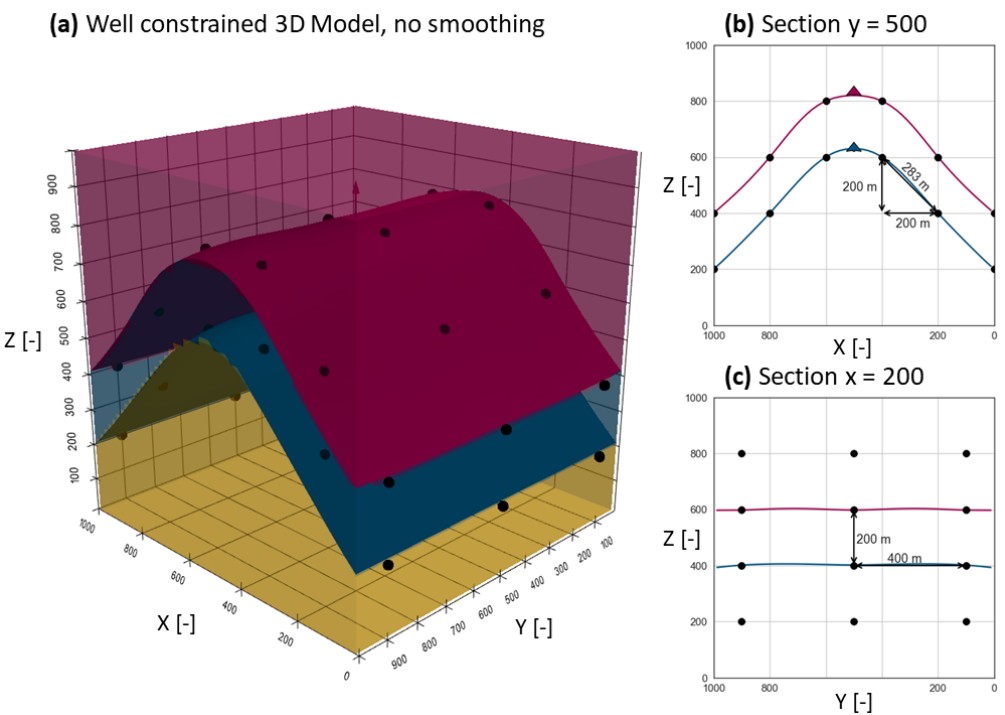

**Figure 5.** Geomodel of a folded unit with regularly spaced input data. 36 surface data points and 2 orientations constrain the model. (**a**) 3D model view. (**b**) Section at y = 500. (**c**) Section at x = 200. Note that input data are projected on section planes.

3.1.1. Model with Random Noise

Assume the model in Figure 5 to be our underlying truth. We now add additional surface points with random noise to the upper surface on both fold limbs. Both limbs can be approached by a geometric plane of the simple form:

$$f(x,y) = 400 + x, \text{for } x \in [0, 400], y \in [0, 1000]$$
$$f(x,y) = 1400 - x, \text{for } x \in [600, 1000], y \in [0, 1000]$$

(7)

Twenty-five additional data surface points at random x and y coordinates were added to each fold limb, with a standard deviation in depth (z) of $\sigma = \pm 25$. The resulting model computed with zero smoothing is shown in Figure 6. We can see that while one fold limb is modeled adequately, showing only small bumps to honor noisy data, the other limb features a geologically unreasonable artifact, where the interpolated surface opens towards the model boundary in a circular shape.

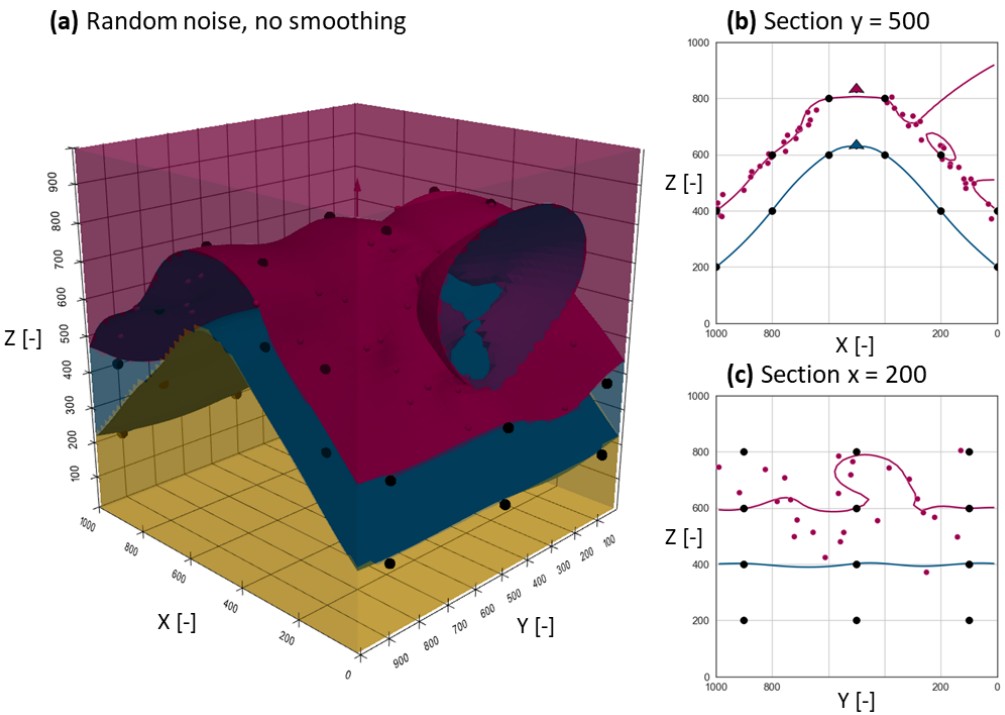

**Figure 6.** Geomodel of a folded unit. 50 noisy data points were added to the data of model in Figure 5. No smoothing was applied. (**a**) 3D model view. (**b**) Section at y = 500. (**c**) Section at x = 200. Note that input data are projected on section planes.

Given that the measurement error in our input data can be quantified and is known for all measurement types, we can apply local smoothing. Assuming that our 36 original surface points are exact and our added surface data points have a known standard deviation of 25 units, we can apply corresponding local smoothing values. In this case, the mapping between scalar field and real space was conducted on the well constrained model (compare Figure 5), leading to a local smoothing of 0.54 for the added surface points. Results for this model can be seen in Figure 7. While small bumps in the interpolated surface exist, undesirable artifacts are removed and the dominant structure is modeled reasonably (compare Figures 5 and 6). Note also that original surface data points are exactly honored, as those feature a local smoothing of zero (e.g., originate from a more reliable source/measurement).

### 3.1.2. Model with Clustered Random Noise

A third model was created to illustrate the effect of locally clustered data. Two point clusters of 25 additional surface points each were added. The first cluster is located around $(400, 900, 800)$, the second one around $(200, 500, 600)$, both with a standard deviation of 25 units in all three directions. The resulting model without smoothing is shown in Figure 8. The resulting artifacts are highly localized compared to the model shown in Figure 6, but still strongly deviate from the desired result, forming pronounced bumps that exceed the input variance as well as unrealistic holes in the resulting surface.

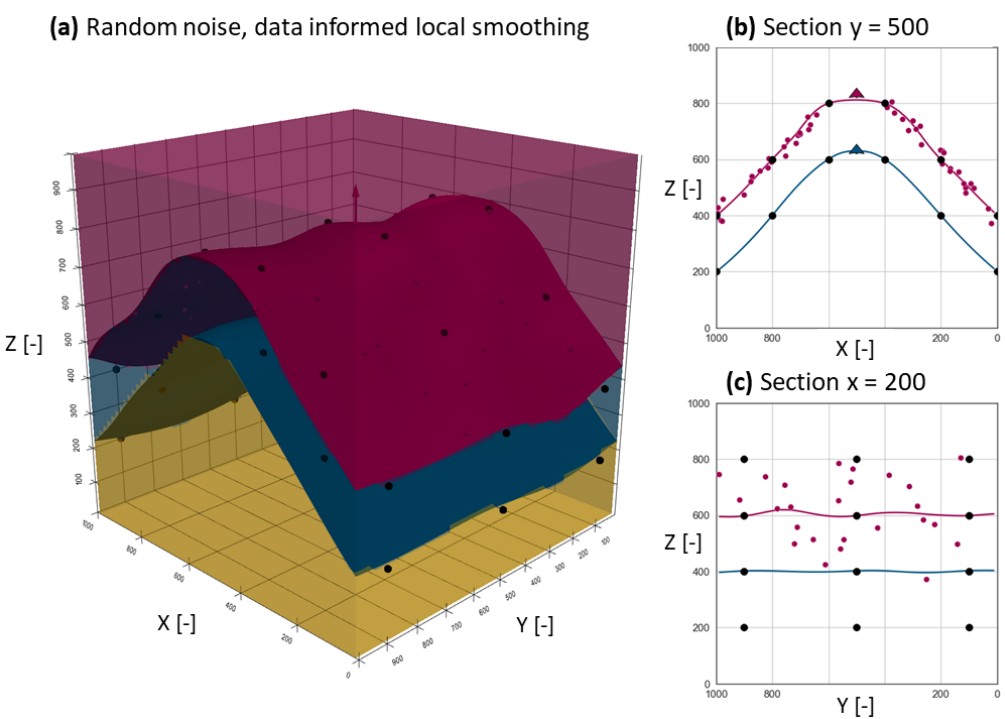

**Figure 7.** Geomodel of a folded unit. Fifty noisy data points were added to the data of model in Figure 5. Original surface data points feature local smoothing of zero, added noisy data points have a local smoothing of 0.54 (standard deviation of 25 scaled to scalar field). (**a**) 3D model view. (**b**) Section viewing at y = 500. (**c**) Section at x = 200. Note that input data are projected on section planes.

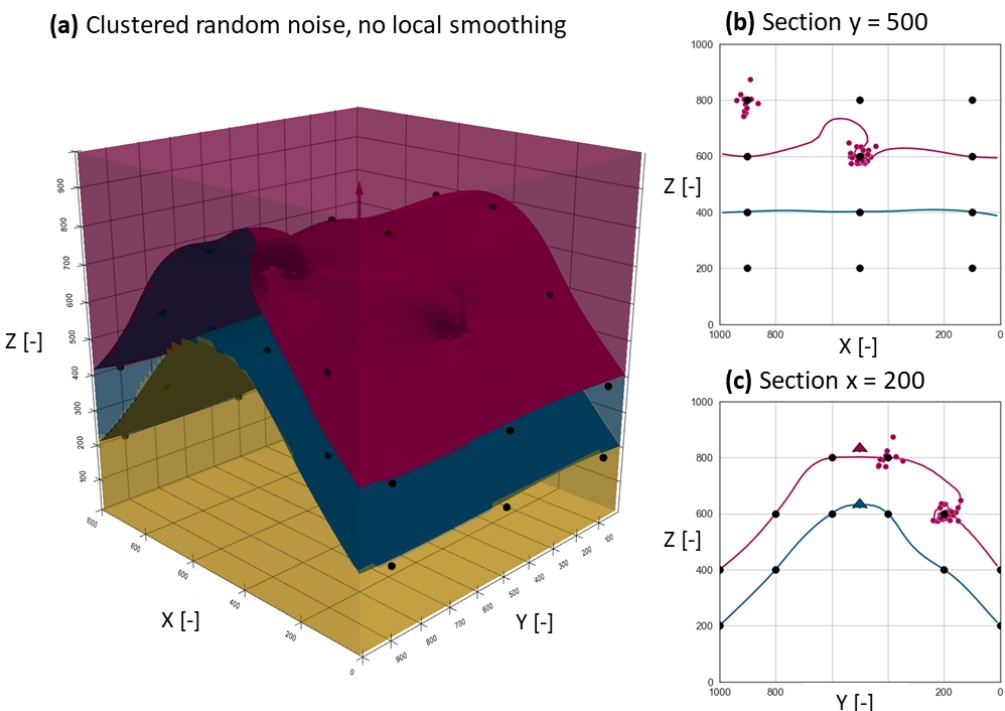

**Figure 8.** Geomodel of a folded unit. Two clusters of 25 noisy data points were added to the data from model in Figure 5. Original surface data points feature local smoothing of zero, added noisy data points have a local smoothing based on KDE. (**a**) 3D model view. (**b**) Section viewing at y = 500. (**c**) Section at x = 200. Note that input data are projected on section planes.

Local smoothing for this model was automatically defined based on the data configuration using Kernel density estimation. The KDE for the given data configuration was calculated using a Gaussian kernel and bandwidth selection following Scott's Rule [39,41]. Resulting KDE values were normalized and can be seen in Figure 9a,b; sections of the resulting locally smoothed model are shown in Figure 9c,d. Point clusters feature high KDE values, leading to smoothing of model artifacts compared to Figure 8, while regularly spaced isolated points are exactly honored, featuring low KDE values.

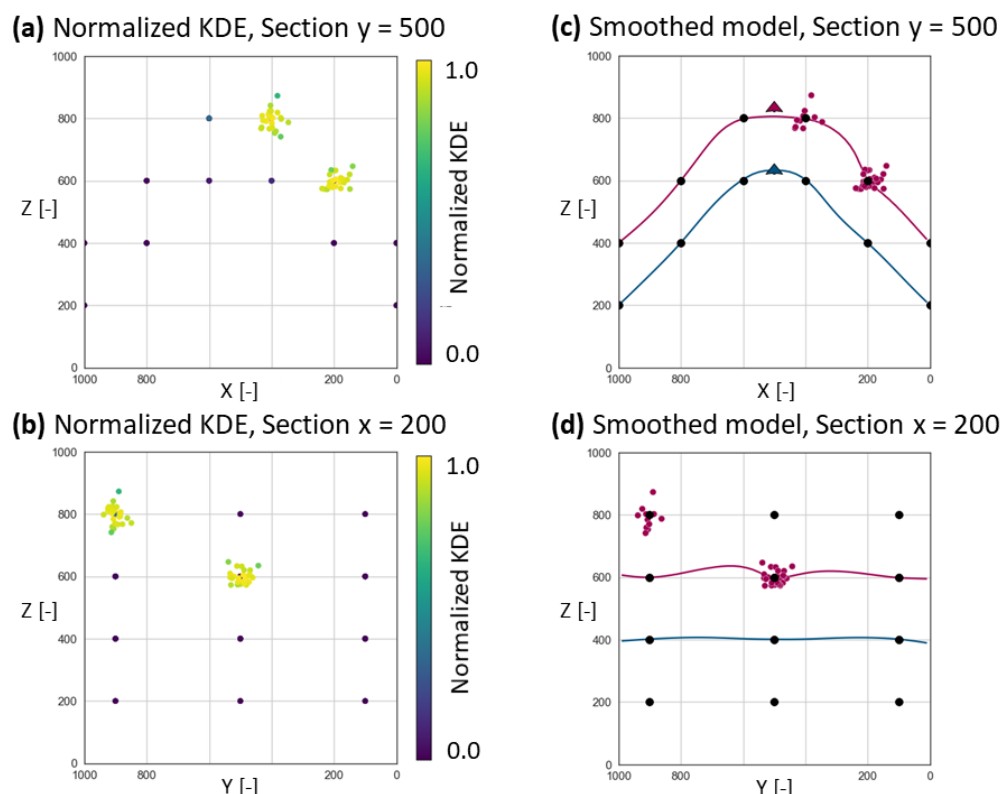

**Figure 9.** Geomodel of a folded unit. Two clusters of 25 noisy data points were added to the data from model in Figure 5. (**a**,**b**) Sections with normalized KDE of surface point input data. (**c**,**d**) Sections of resulting model with KDE informed local smoothing. Note that input data are projected on section planes.

## 4. Discussion

Herein, we present a method to allow interpolated surfaces in geomodeling with the potential field method to deviate from given input surface data points based on a single parameter for each input data point. In this scenario, the kriging-based potential field method ceases to be an exact interpolator. In a purely mathematical sense we therefore decrease the accuracy of our model. We still consider the presented method to be of important practical use in two scenarios: The straightforward case, that directly corresponds to the concept of filtered kriging [18,34,35], is when a quantified measurement error or uncertainty for surface data points is available. The second case, more specific to geologic modeling, is a question of scale: especially when we have irregular data configuration, small-scale variation might be captured by these measurements in some parts of the model, while not at all in others. This becomes a problem when it negatively affects model robustness and leads to modeling artifacts. Local smoothing provides a tool for practitioners to create more robust models on the required scale fit for their purpose [2].

In case of known uncertainties, the method enables the integration of data from various sources in a single framework, reducing the amount of manual data selection that often leads to irreproducible results. If uncertainties are unknown, kernel density-informed local

smoothing can be used to avoid artifacts that result from a irregular and clustered, data configuration. The method is easy to implement and the underlying fundamental concepts are well described in the traditional kriging literature [18,34,35]. A current limitation is the scaling of the local smoothing value relative to the scalar field. For low complexity models, a mapping between scalar field and real space is straightforward to establish. With increasing model complexity, this mapping becomes more difficult to obtain and will require additional steps in the modeling workflow (compare Section 2.6). We acknowledge that tuning this parameter is comparable to adjusting the covariance structure, but it also implicitly covers typical data preprocessing steps (data selection) and reduces the amount of necessary postprocessing (compare Section 1). This reduces the complexity, increases the reproducibility and simplifies optimization of workflows.

We want to emphasize that the suggested local smoothing parameter can be integrated into a geophysical inversion framework. Optimizing for the local smoothing as a model parameter will solve the scaling issue and help to create robust inversion results [43].

Other approximation methods for fitting scalar field interpolants can be used to reduce the occurrence and magnitude of modeling artifacts resulting from fitting noisy and clustered datasets. Three such methods include the use of greedy algorithms [44], inequality constraints [45] and convolving an interpolant with a smoothing kernel [46]. Greedy algorithms start fitting the interpolant to a minimal subset of the dataset. The interpolant is evaluated on data points there we were not included in this subset to compute errors between the data and the current interpolant. The largest errors beyond some user-defined threshold are added to a refitted interpolant. The algorithm stops when there are no error beyond this threshold. Inequality constraints can be used to define ranges of scalar values and orientation using known data uncertainty that the smoothest surface solution be found within. Last, smoothing kernels provide a method to filter out high frequency geometrical variations by switching the kernels used to fit the interpolant with their associated smoothing kernels. Degree of smoothing is adjusted by changing the parameters of the smoothing kernels without requiring the interpolant to be refitted.

The strength of the local smoothing method in comparison to these approximation methods from the field of computer graphics are the straightforward integration into existing geomodeling software, the available amount of background literature for kriging-based methods, full manual control (compare Section 2.5.1), a direct and easy-to-comprehend link between local smoothing and typical multi-source geoscientific input datasets (compare Section 2.5.2), as well as fully automated smoothing for clustered points (compare Section 2.5.3). The approach is versatile and a combination of information methods for the local smoothing can be combined for complex datasets.

## 5. Conclusions

In this work, we address the problem of modeling artifacts in implicit 3D geomodeling caused by data configuration, data quality and scale-dependent model variability using a single local smoothing parameter per input point that can be informed in various ways.

While the underlying theory of the proposed approach is well known from classical geostatistics, the application to 3D geomodeling with the potential field method is innovative and useful as practitioners are given a tool to reduce geologically unreasonable modeling artifacts. Adjusting a single parameter, instead of a using complex mix of interacting approaches, improves usability and reproducibility. Local smoothing allows accounting for artifacts caused by scale variation and measurement uncertainty, but also offers enhanced manual control if required. The main limitation at the current state is the scaling of the parameter to the dimensionless scalar field. We want to emphasize that application and value of the method depend on the structure and quality of available datasets and the defined model purpose.

Future research should focus on additional sources to inform the local smoothing parameter, fully automated scaling of these values to the scalar field, as well as application in case studies using real datasets. The reduction of artifacts also promotes the use of

the method in cases where multiple model realizations are required, for example, in the growing field of uncertainty quantification.

**Author Contributions:** Conceptualization, J.v.H. and M.d.l.V.; methodology, J.v.H.; software, J.v.H. and M.d.l.V.; formal analysis, J.v.H.; investigation, J.v.H.; writing—original draft preparation, J.v.H.; writing—review and editing, F.W. and M.H.; visualization, J.v.H.; supervision, F.W.; project administration, F.W.; funding acquisition, M.d.l.V. and F.W. All authors have read and agreed to the published version of the manuscript.

**Funding:** This activity has received funding from the European Institute of Innovation and Technology (EIT), project number 19004 (FARMIN). This body of the European Union receives support from the European Union's Horizon 2020 and Horizon Europe research and innovation programmes.

**Institutional Review Board Statement:** Not applicable

**Data Availability Statement:** The potential field method is implemented in the open-source geomodeling package GemPy (https://www.gempy.org/ (accessed on 8 November 2021) and https://github.com/cgre-aachen/gempy (accessed on 8 November 2021)). Additional material to support the method proposed in this work can be found under https://github.com/cgre-aachen/InformedLocalSmoothing (accessed on 8 November 2021).

**Conflicts of Interest:** The authors declare no conflict of interest.

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
