# Peer review of "Informed Local Smoothing in 3D Implicit Geological Modeling"

_minerals, doi:10.3390/min11111281_

Round 1

Reviewer 1 Report

I provide my comments below for the authors to consider for further improve the quality of the manuscript.

  • It is not clear to me what would be the innovation and original contribution to the current knowledge, particularly, in comparison with the existing literature, and I suggest the authors have to emphasize this matter clearly.
    I suggest the authors include a flowchart describing the main steps to implement the model for the benefit of readers.
  • Figures 8 & 9 are not visible, I can hardly see the text, also the font size of the figure caption is very small.
  • I suggest including a Conclusion section covering key findings/outcomes from the current study.

Author Response

Response to reviewer 1 comments

I provide my comments below for the authors to consider for further improve the quality of the manuscript.

Thank you for the suggestions and the helpful comments. The following changes were made to improve the manuscript accordingly:

It is not clear to me what would be the innovation and original contribution to the current knowledge, particularly, in comparison with the existing literature, and I suggest the authors have to emphasize this matter clearly. I suggest the authors include a flowchart describing the main steps to implement the model for the benefit of readers. I suggest including a Conclusion section covering key findings/outcomes from the current study.

A conclusion section was added (ll. 420-439) to summarize the key findings and advantages, as well as to give an outlook on future research. The difference between the approach in classical geostatistics and its application in 3D geomodeling was emphasized in the introduction (ll. 123-128)  and conclusion to highlight the innovation and original contribution of this work. We refrained from adding a flow chart explaining the implementation steps, as we feel this is adequately covered by the provided equations and the supporting code provided under “data availability” (ll. 449-452).

Figures 8 & 9 are not visible, I can hardly see the text, also the font size of the figure caption is very small.

All figures in the results section were revised to improve readability. Overall size and label sizes were increased. 

We believe these adjustments improve the quality of the manuscript and thank you for the valuable help and insight. Please find a revised manuscript with marked changes attached.

Reviewer 2 Report

The manuscript proposes to use the local smoothing method in comparison to standard kriging, the nugglet effect and the potential field method in 3D geological modeling. The considered approaches are illustrated using a fairly simple model. It would be interesting to add a more complex, realistic model to see how the different methods behave. 

Author Response

Response to reviewer 2 comments

The manuscript proposes to use the local smoothing method in comparison to standard kriging, the nugglet effect and the potential field method in 3D geological modeling. The considered approaches are illustrated using a fairly simple model. It would be interesting to add a more complex, realistic model to see how the different methods behave. 

We thank the reviewer for taking the time to review this submission. We are grateful for the helpful comments and suggestions. In this manuscript we tried to focus purely on the mathematical background and therefore used simple examples to highlight the essential possibilities of the method. We agree that a real case study will be interesting and we strive to provide such a study in a future publication.

Please find a revised version of the manuscript with marked changes attached.

Reviewer 3 Report

The authors presented an informed local smoothing approach for representation of geological geological structures in 3D. The paper is well written, however I have some suggestions for possible improvements

1. Why a more complex geological model was not demonstrated? 
2. The authors claimed that existing methods have well-known problems with inhomogeneous data distributions that this method is trying to fix. On the other hand, the solution presented still requires a careful choice of parameter settings for such datasets. Is this really an improvement? Is it not possible to generate similar results with existing methods with a careful choice of parameter settings?
3. A "Conclusions" section seems to be missing. 

Author Response

Response to reviewer 3 comments

The authors presented an informed local smoothing approach for representation of geological geological structures in 3D. The paper is well written, however I have some suggestions for possible improvements

Thank you for the suggestions and the helpful comments. The following changes were made to improve the manuscript accordingly:

  1. Why a more complex geological model was not demonstrated? 

In this manuscript we tried to focus purely on the mathematical background and hope that the provided examples highlight the essence of the method. We agree that a real world case study would be an important addition and we strive to provide such a study in a future publication.

  1. The authors claimed that existing methods have well-known problems with inhomogeneous data distributions that this method is trying to fix. On the other hand, the solution presented still requires a careful choice of parameter settings for such datasets. Is this really an improvement? Is it not possible to generate similar results with existing methods with a careful choice of parameter settings?

We added a paragraph to compare the parameter tuning to the other presented approaches that aim to reduce modeling artefacts and clarified where we see the major advantages (ll. 388-392).

  1. A "Conclusions" section seems to be missing. 

A conclusion section was added (ll. 420-439) to summarize the key findings and advantages, as well as to give an outlook on future research. 

We believe these adjustments improve the quality of the manuscript and thank you for the valuable help and insight. Please find a revised version of the manuscript with marked changes attached.

Reviewer 4 Report

This paper presents an approach to overcome the geological model's ordinary method deficiency through a combination of an implicit interpolation algorithm with a local smoothing approach based on the concepts of nugget effect and filtered kriging known from conventional geostatistics. The paper reads well and can be accepted after minor revision. See the below:

  1. There are couple of statistical techniques in the method section but it requires to explain the advantages and disadvantages/limitations of those statistical techniques for example, kriging, kernel density estimation and so on.
  2. I would recommend to provide a conclusion section by mentioning the key findings, future direction of research etc.

Author Response

Response to reviewer 4 comments

This paper presents an approach to overcome the geological model's ordinary method deficiency through a combination of an implicit interpolation algorithm with a local smoothing approach based on the concepts of nugget effect and filtered kriging known from conventional geostatistics. The paper reads well and can be accepted after minor revision. See the below:

Thank you for taking the time to review this submission. We are grateful for the helpful comments and suggestions. The following changes were made to improve the manuscript accordingly: 

  • There are couple of statistical techniques in the method section but it requires to explain the advantages and disadvantages/limitations of those statistical techniques for example, kriging, kernel density estimation and so on.

We improved the description of the statistical techniques in the method section (ll. 186-190 and ll. 287-291), trying to focus on advantages and limitations of each technique

  • I would recommend to provide a conclusion section by mentioning the key findings, future direction of research etc.

A conclusion section was added (ll. 420-439) to summarize the key findings and advantages, as well as to give an outlook on future research. 

We believe these adjustments improve the quality of the paper and thank you for the valuable help and insight. Please find a revised version of the manuscript with marked changes attached.

Reviewer 5 Report

First of all, I would like to thank the Editorial Board of the Minerals Journal for giving me the opportunity to participate in the review process of this paper.

The paper deals with a topic of undoubted interest (although perhaps it could have a better audience in other types of journals) such as the 3D modeling of geological phenomena. The contribution focuses on a procedure to try to avoid the problems derived from the application of modeling techniques (in particular, geostatistical modeling) to real data. This is undoubtedly an interesting aspect, since in reality and, on many occasions, what we are facing is a problem of 5 dimensions, that to the traditional ones of the geometric position of the data (XYZ), we must add the temporal variable (time -t-) basic in the analysis of phenomena with temporal evolution (or even within the mining operations themselves in the evolution of the mining cuts), and finally, consider the variable related to the uncertainty of the information that we should consider as a fifth dimension to be considered.

However, in my opinion, the paper presented does not clarify the solutions proposed, and those proposed from my point of view do not have the appropriate level of scientific rigor required. Also, it would have been of interest to analyze a set of real data, since the examples presented cannot be considered as representative of real problems that may be of interest to the readers of this journal.

As a basic element, the modification of the parameters derived from the structural analysis of the experimental data (which can certainly come from various sources) is proposed in an artificial way to eliminate the "artifacts" that can occur in the geostatistical estimation processes by means of kriging. In this regard, it is important to keep in mind that the theoretical foundation of kriging -and Geostatistics through the Regionalized Variables Theory- is centered precisely on this, on starting from the available experimental information, in order to improve the estimation procedures through the process of structural analysis.

The appearance of these artifacts may be due to problems in our starting information or to poor choices in the estimation parameters, such as, for example, anisotropies and search radii that are not well adjusted to the ranges of the semivariograms, number of points involved in the estimation, etc. It is also clear that other methods can be used to consider these uncertainties, but I am never in favor of reducing them, but of considering them (using for example geostatistical simulations) because they ultimately reflect the reality of our data, at the level of experimental data, or of the behavior of the variable under consideration (e.g., layer thickness).

In this sense, I do not consider that the modification of these parameters to others from those obtained from the experimental information itself is adequate, although it may serve to obtain "smoothed" models. 

For these reasons, I consider that this publication is not suitable for publication in the Minerals journal. Therefore, I recommend to its authors a thorough revision of it, and if necessary, the resubmission to the journal of the new version.

Author Response

Response to reviewer 5 comments

First of all, I would like to thank the Editorial Board of the Minerals Journal for giving me the opportunity to participate in the review process of this paper.

The paper deals with a topic of undoubted interest (although perhaps it could have a better audience in other types of journals) such as the 3D modeling of geological phenomena. The contribution focuses on a procedure to try to avoid the problems derived from the application of modeling techniques (in particular, geostatistical modeling) to real data. This is undoubtedly an interesting aspect, since in reality and, on many occasions, what we are facing is a problem of 5 dimensions, that to the traditional ones of the geometric position of the data (XYZ), we must add the temporal variable (time -t-) basic in the analysis of phenomena with temporal evolution (or even within the mining operations themselves in the evolution of the mining cuts), and finally, consider the variable related to the uncertainty of the information that we should consider as a fifth dimension to be considered.

Thank you for taking the time to review this submission and for your insightful comments, especially concerning the fundamental geostatistical aspects. Our aim with this contribution is, in fact, to contribute to the practical application of geostatistical methods in the structural modeling of crustal structures and mineral deposits - hence also our choice to submit it to the special issue. And as we agree that data should not be removed, but instead, a suitable strategy to adjust the interpolation on the basis of (locally) noisy data is promoted here. 

However, in my opinion, the paper presented does not clarify the solutions proposed, and those proposed from my point of view do not have the appropriate level of scientific rigor required. Also, it would have been of interest to analyze a set of real data, since the examples presented cannot be considered as representative of real problems that may be of interest to the readers of this journal.

In this manuscript we tried to focus purely on the mathematical background and hope that the provided examples highlight the essence of the method. We agree that a real world case study would be an important addition and we strive to provide such a study in a future publication.

As a basic element, the modification of the parameters derived from the structural analysis of the experimental data (which can certainly come from various sources) is proposed in an artificial way to eliminate the "artifacts" that can occur in the geostatistical estimation processes by means of kriging. In this regard, it is important to keep in mind that the theoretical foundation of kriging -and Geostatistics through the Regionalized Variables Theory- is centered precisely on this, on starting from the available experimental information, in order to improve the estimation procedures through the process of structural analysis.

The appearance of these artifacts may be due to problems in our starting information or to poor choices in the estimation parameters, such as, for example, anisotropies and search radii that are not well adjusted to the ranges of the semivariograms, number of points involved in the estimation, etc. It is also clear that other methods can be used to consider these uncertainties, but I am never in favor of reducing them, but of considering them (using for example geostatistical simulations) because they ultimately reflect the reality of our data, at the level of experimental data, or of the behavior of the variable under consideration (e.g., layer thickness).

In our revisions we tried to emphasize the inherent difference of classical geostatistical applications, where we agree that smoothing is not generally favorable, to geomodeling, where a robust and geologically reasonable model is often the main focus (ll. 420-439). In this context, traditional simulations are often not feasible, specifically because of the modeling artefacts described in our work. Semivariograms and search radii are similarly difficult to define for structural modeling and are often chosen arbitrarily (featuring very high ranges) in practice. We agree that a proper integration of anisotropies can improve modeling results and hope to work on this in the future.

In this sense, I do not consider that the modification of these parameters to others from those obtained from the experimental information itself is adequate, although it may serve to obtain "smoothed" models. 

For these reasons, I consider that this publication is not suitable for publication in the Minerals journal. Therefore, I recommend to its authors a thorough revision of it, and if necessary, the resubmission to the journal of the new version.

We believe the adjustments made based on the review improve the quality of the paper and thank you for the valuable help and insight. Please find a revised version of the manuscript with marked changes attached.

Round 2

Reviewer 1 Report

The Authors have satisfactorily attended my comments and suggestion. I have now recommended for the publication of the revised manuscript in the Minerals Journal.

Author Response

Thank your taking the time to review this submission. We appreciate the helpful suggestions and the positive feedback.

Reviewer 2 Report

Unfortunately, my comment was ignored. The point is that the manuscript states the following: «we want to emphasize that application and value of the method depend on the structure and quality of available datasets and the defined model purpose». Thus, it is completely unclear how the proposed method will behave in a real situation. It would be possible to complicate the models in some way and see how the smoothing result changes. I suggest rejecting the manuscript until this is done.

Author Response

Unfortunately, my comment was ignored. The point is that the manuscript states the following: «we want to emphasize that application and value of the method depend on the structure and quality of available datasets and the defined model purpose». Thus, it is completely unclear how the proposed method will behave in a real situation. It would be possible to complicate the models in some way and see how the smoothing result changes. I suggest rejecting the manuscript until this is done.

We are sorry to hear that you feel your comment was not acknowledged. A complete case study, however, would exceed the purpose and scope of this submission. We do not feel that increased complexity using artificial models and noise data would add value to the manuscript, as the effect of the method is showcased well by the given examples. Thank you for taking the time to review this submission and for your comments and suggestions.

Reviewer 5 Report

Having reviewed the new version of the manuscript, in my opinion it does not include adequate responses to the problems that were pointed out in my first review. In many cases, the authors, according to their response letter, agree with the interest of the same but indicate that these aspects will be the subject of future research (and publications).

In this sense, even the comments incorporated into the work raise important doubts in this regard. For example:

In L123 it does not make clear what the real objective is, to obtain smoothed models? models that fit reality, which by the way, is not usually excessively continuous on many occasions? to incorporate the uncertainties of the different measurement sets, although in this case, what is proposed is directly a smoothing to eliminate the local variability detected, instead of considering it as an element of representation of the phenomenon?

L189 states that kriging ceases to be an exact interpolator, something that can never be considered as true. Kriging acts according to the available information (data+spatial variability structures) and the estimation results are adjusted to it, another problem may be that the available information is insufficient or inadequate.

L388 it is recognized that this implies a change in the covariance functions (not only in some cases, but whenever adjustments are incorporated in the weights according to the geostatistical methodology), which is only justified from the point of view of complexity reduction but not of the quality of the information provided by the models themselves.

In this sense, I can only ratify my previous decision to consider that this work does not meet the requirements for publication in a journal of the quality level of Minerals.

Author Response

Having reviewed the new version of the manuscript, in my opinion it does not include adequate responses to the problems that were pointed out in my first review. In many cases, the authors, according to their response letter, agree with the interest of the same but indicate that these aspects will be the subject of future research (and publications).

Thank you for reviewing the revised manuscript. We appreciate the thorough review and the helpful comments and suggestions.

In this sense, even the comments incorporated into the work raise important doubts in this regard. For example:

In L123 it does not make clear what the real objective is, to obtain smoothed models? models that fit reality, which by the way, is not usually excessively continuous on many occasions? to incorporate the uncertainties of the different measurement sets, although in this case, what is proposed is directly a smoothing to eliminate the local variability detected, instead of considering it as an element of representation of the phenomenon?

The point described here puts focus on a very important distinction: From a traditional geostatistical perspective the proposed smoothing is not in general desireable. Local variability is exactly what we try to capture when applying kriging interpolation (or simulation) to model phenomenons like ore grade or porosity distributions. 3D structural modeling though is a different field. While the potential field method is based on geostatistical principles, the main objective is to create a structural model that is based on data, but also honors geologic knowledge and logic. Clearly unrealistic modeling artefacts, like the ones shown in this manuscript (compare figures 6 and 8), can occur purely based on data configuration, measurement errors and scale dependency and do not necessarily bear any geologic meaning. While this can be seen as a weakness of the implicit potential field approach itself, it is still applied widely in 3D structural modeling. As described in our introduction, alternative methods to avoid these artefacts often include tedious and irreproducible manual work. The value of our work therefore lies in the application of local smoothing specifically in 3D structural modeling, where it is often not possible or feasible to “consider local variability as an element of representation of the phenomenon” due to the specific modeling approach, geologic reason, model scale or model purpose.

L189 states that kriging ceases to be an exact interpolator, something that can never be considered as true. Kriging acts according to the available information (data+spatial variability structures) and the estimation results are adjusted to it, another problem may be that the available information is insufficient or inadequate.

We use the term “exact” in this context in accordance with kriging literature (compare for example Wackernagel (2003), p.81 or Webster & Oliver (2007), p. 158) to describe a characteristic of the kriging interpolator. Following this nomenclature, kriging is an exact interpolator in absence of a nugget effect, as the interpolation will always equal the measurements at observed locations. By applying a global nugget effect, or, as in our case, a local smoothing parameter based on the principle of the nugget effect, the interpolator loses this characteristic and ceases to be “exact” (compare for example Webster & Oliver (2007), p. 180). Kriging, as an interpolator, is of course still deterministic, and thus acts according to the available information, as correctly pointed out. This is also a very important feature in the context of our work, as it enables reproducibility, which is often omitted in practical applications of geological modeling. We revised the section to clarify the meaning of the term “exact” in the given context (ll. 189-191).

L388 it is recognized that this implies a change in the covariance functions (not only in some cases, but whenever adjustments are incorporated in the weights according to the geostatistical methodology), which is only justified from the point of view of complexity reduction but not of the quality of the information provided by the models themselves.

The method we suggest is equivalent to locally changing the measurement error related portion of the nugget effect. Therefore we do agree that this can always be viewed as a local change in the covariance function (adapted in ll. 389-2), as the nugget effect is part of this. As pointed out above, we do strongly believe that a targeted (local) and reproducible reduction of variability (not complexity, as we do not reduce the number of model parameters) is a valuable addition to a toolbox of 3D structural modeling techniques. The studies referenced in the introduction (ll. 56-59) and the presented examples are meant to emphasize this relevance.

In this sense, I can only ratify my previous decision to consider that this work does not meet the requirements for publication in a journal of the quality level of Minerals.

We are sorry that you disagree with the relevance of the suggested method and conducted work - this is certainly a subjective consideration. However, the approach that we have taken follows a clear scientific structure and we think that it fits into the scope of the journal itself, and especially into the special issue to which it was submitted. 

We hope the revisions and explanations above help to clarify some important points and thank you again for taking the time to review the manuscript and for providing valuable insight and perspective.

References

Wackernagel, H. Multivariate Geostatistics: An Introduction with Applications, third, completely revised edition ed.; Springer: Berlin and Heidelberg, 2003. 

Webster, R.; Oliver, M.A. Geostatistics for environmental scientists, 2nd ed. ed.; Statistics in practice, Wiley: Chichester, 2007.